# Edge Guided GANs with Contrastive Learning for Semantic Image Synthesis

**Hao Tang[1], Xiaojuan Qi[2], Guolei Sun[1], Dan Xu[3], Nicu Sebe[4], Radu Timofte[1,5], Luc Van Gool[1]**
[1]ETH Zurich, [2]University of Hong Kong, [3]HKUST, [4]University of Trento, [5]University of Wurzburg

## Abstract

We propose a novel edge guided generative adversarial network with contrastive learning (ECGAN) for the challenging semantic image synthesis task. Although considerable improvement has been achieved, the quality of synthesized images is far from satisfactory due to three largely unresolved challenges. 1) The semantic labels do not provide detailed structural information, making it difficult to synthesize local details and structures. 2) The widely adopted CNN operations such as convolution, down-sampling, and normalization usually cause spatial resolution loss and thus cannot fully preserve the original semantic information, leading to semantically inconsistent results (e.g., missing small objects). 3) Existing semantic image synthesis methods focus on modeling "local" semantic information from a single input semantic layout. However, they ignore "global" semantic information of multiple input semantic layouts, i.e., semantic cross-relations between pixels across different input layouts. To tackle 1), we propose to use edge as an intermediate representation which is further adopted to guide image generation via a proposed attention guided edge transfer module. Edge information is produced by a convolutional generator and introduces detailed structure information. To tackle 2), we design an effective module to selectively highlight class-dependent feature maps according to the original semantic layout to preserve the semantic information. To tackle 3), inspired by current methods in contrastive learning, we propose a novel contrastive learning method, which aims to enforce pixel embeddings belonging to the same semantic class to generate more similar image content than those from different classes. Doing so can capture more semantic relations by explicitly exploring the structures of labeled pixels from multiple input semantic layouts. Experiments on three challenging datasets show that our ECGAN achieves significantly better results than state-of-the-art methods.

## 1 Introduction

Semantic image synthesis refers to generating photo-realistic images conditioned on pixel-level semantic labels. This task has a wide range of applications such as image editing and content generation (Chen & Koltun, 2017; Isola et al., 2017; Guo et al., 2022; Gu et al., 2019; Bau et al., 2019a;b; Liu et al., 2019; Qi et al., 2018; Jiang et al., 2020). Although existing methods conducted interesting explorations, we still observe unsatisfactory aspects, mainly in the generated local structures and details, as well as small-scale objects, which we believe are mainly due to three reasons: 1) Conventional methods (Park et al., 2019; Wang et al., 2018; Liu et al., 2019) generally take the semantic label map as input directly. However, the input label map provides only structural information between different semantic-class regions and does not contain any structural information within each semantic-class region, making it difficult to synthese rich local structures within each class. Taking label map $S$ in Figure 1 as an example, the generator does not have enough structural guidance to produce a realistic bed, window, and curtain from only the input label ($S$). 2) The classic deep network architectures are constructed by stacking convolutional, down-sampling, normalization, non-linearity, and up-sampling layers, which will cause the problem of spatial resolution losses of the input semantic labels. 3) Existing methods for this task are typically based on global image-level generation. In other words, they accept a semantic layout containing several object classes and aim to generate the appearance of each one using the same network. In this way, all the classes are

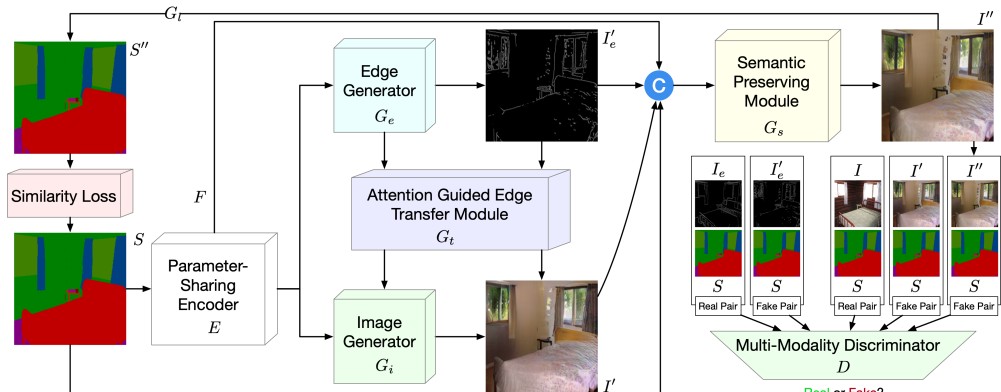

Figure 1: Overview of the proposed ECGAN. It consists of a parameter-sharing encoder $E$, an edge generator $G_e$, an image generator $G_i$, an attention guided edge transfer module $G_t$, a label generator $G_l$, a similarity loss module, a contrastive learning module $G_c$ (not shown for brevity), and a multi-modality discriminator $D$. $G_e$ and $G_i$ are connected by $G_t$ from two levels, i.e., edge feature-level and content-level, to generate realistic images. $G_s$ is proposed to preserve the semantic information of the input semantic labels. $G_l$ aims to transfer the generated image back to the label for calculating the similarity loss. $G_c$ tries to capture more semantic relations by explicitly exploring the structures of labeled pixels from multiple input semantic layouts. $D$ aims to distinguish the outputs from two modalities, i.e., edge and image. The symbol ⓒ denotes channel-wise concatenation.

treated equally. However, because different semantic classes have distinct properties, using specified network learning for each would intuitively facilitate the complex generation of multiple classes.

To address these three issues, in this paper, we propose a novel edge guided generative adversarial network with contrastive learning (ECGAN) for semantic image synthesis. The overall framework of the proposed ECGAN is shown in Figure 1.

To tackle 1), we first propose an edge generator to produce the edge features and edge maps. Then the generated edge features and edge maps are selectively transferred to the image generator and improve the quality of the synthesized image by using our attention guided edge transfer module.

To tackle 2), we propose an effective semantic preserving module, which aims at selectively highlighting class-dependent feature maps according to the original semantic layout. We also propose a new similarity loss to model the relationship between semantic categories. Specifically, given a generated label $S''$ and corresponding ground truth $S$, similarity loss constructs a similarity map to supervise the learning.

To tackle 3), a straightforward solution would be to model the generation of different image classes individually. By so doing, each class could have its own generation network structure or parameters, thus greatly avoiding the learning of a biased generation space. However, there is a fatal disadvantage to this. That is, the number of parameters of the network will increase linearly with the number of semantic classes $N$, which will cause memory overflow and make it impossible to train the model. If we use $p_e$ and $p_d$ to denote the number of parameters of the encoder and decoder, respectively, then the total number of the network parameter should be $p_e + N \times p_d$ since we need a new decoder for each class. To further address this limitation, we introduce a pixel-wise contrastive learning approach that elevates the current image-wise training method to a pixel-wise method. By leveraging the global semantic similarities present in labeled training layouts, this method leads to the development of a well-structured feature space. In this case, the total number of the network parameter only is $p_e + p_d$. Moreover, we explore image generation from a class-specific context, which is beneficial for generating richer details compared to the existing image-level generation methods. A new class-specific pixel generation strategy is proposed for this purpose. It can effectively handle the generation of small objects and details, which are common difficulties encountered by the global-based generation.

With the proposed ECGAN, we achieve new state-of-the-art results on Cityscapes (Cordts et al., 2016), ADE20K (Zhou et al., 2017), and COCO-Stuff (Caesar et al., 2018) datasets, demonstrating the effectiveness of our approach in generating images with complex scenes, and showing significantly better results compared with state-of-the-art methods.

## 2 RELATED WORK

**Edge Guided Image Generation.** Edge maps are usually adopted in image inpainting (Ren et al., 2019; Nazeri et al., 2019a; Li et al., 2019) and image super-resolution (Nazeri et al., 2019b) tasks to reconstruct the missing structure information of the inputs. For example, Nazeri et al. (2019a) proposed an edge generator to hallucinate edges in the missing regions given edges, which can be regarded as an edge completion problem. Using edge images as the structural guidance, EdgeConnect (Nazeri et al., 2019a) achieves good results even for some highly structured scenes. Unlike previous works, including EdgeConnect, we propose a novel edge generator to perform a new task, i.e., semantic label-to-edge translation. To the best of our knowledge, we are the first to generate edge maps from semantic labels. Then the generated edge maps, with more local structure information, can be used to improve the quality of the image results.

**Semantic Image Synthesis** aims to generate a photo-realistic image from a semantic label map (Chen & Koltun, 2017; Park et al., 2019; Liu et al., 2019; Bansal et al., 2019; Zhu et al., 2020a; Ntavelis et al., 2020; Zhu et al., 2020b; Sushko et al., 2021; Tan et al., 2021b;a; Zhu et al., 2020b). With semantic information as guidance, existing methods have achieved promising performance. However, we can still observe unsatisfying aspects, especially on the generation of the small-scale objects, which we believe is mainly due to the problem of spatial resolution losses associated with deep network operations such as convolution, normalization, down-sampling, etc. To solve this problem, Park et al. (2019) proposed GauGAN, which uses the input semantic labels to modulate the activations in normalization layers through a spatially-adaptive transformation. However, the spatial resolution losses caused by other operations, such as convolution and down-sampling, have not been resolved. Moreover, we observe that the input label map has only a few semantic classes in the entire dataset. Thus the generator should focus more on learning these existing semantic classes rather than all the semantic classes. To tackle both limitations, we propose a novel semantic preserving module, which aims to selectively highlight class-dependent feature maps according to the input labels for generating semantically consistent images. We also propose a new similarity loss to model the intra-class and inter-class semantic dependencies.

**Contrastive Learning.** Recently, the most compelling methods for learning representations without labels have been unsupervised contrastive learning (Oord et al., 2018; Hjelm et al., 2018; Pan et al., 2021; Wu et al., 2018; Chen et al., 2020), which significantly outperform other pretext task-based alternatives (Gidaris et al., 2018; Doersch et al., 2015; Noroozi & Favaro, 2016). Contrastive learning aims to learn the general features of unlabeled data by teaching and guiding the model which data points are different or similar. For example, Hu et al. (2021) designed a region-aware contrastive learning to explore semantic relations for the specific semantic segmentation problem. Zhao et al. (2021) and Wang et al. (2021) also proposed two new contrastive learning-based strategies for the semantic segmentation task. However, we propose a novel pixel-wise contrastive learning method for semantic image synthesis in this paper. The semantic image synthesis task is very different from the semantic segmentation task, which requires us to tailor the network structure and loss function to our generation task. Specifically, we propose a new training protocol that explores global pixel relations in labeled layouts for regularizing the generation embedding space.

## 3 EDGE GUIDED GANS WITH CONTRASTIVE LEARNING

**Framework Overview.** Figure 1 shows the overall structure of ECGAN for semantic image synthesis, which consists of a semantic and edge guided generator $G$ and a multi-modality discriminator $D$. The generator $G$ consists of eight components: (1) a parameter-sharing convolutional encoder $E$ is proposed to produce deep feature maps $F$; (2) an edge generator $G_e$ is adopted to generate edge maps $I'_e$ taking as input deep features from the encoder; (3) an image generator $G_i$ is used to produce intermediate images $I'$; (4) an attention guided edge transfer module $G_t$ is designed to forward useful structure information from the edge generator to the image generator; (5) the semantic preserving module $G_s$ is developed to selectively highlight class-dependent feature maps according to the input label for generating semantically consistent images $I''$; (6) a label generator $G_l$ is employed to produce the label from $I''$; (7) the similarity loss is proposed to calculate the intra-class and inter-class relationships. (8) the contrastive learning module $G_c$ aims to model global semantic relations between training pixels, guiding pixel embeddings towards cross-image category-discriminative representations that eventually improve the generation performance.

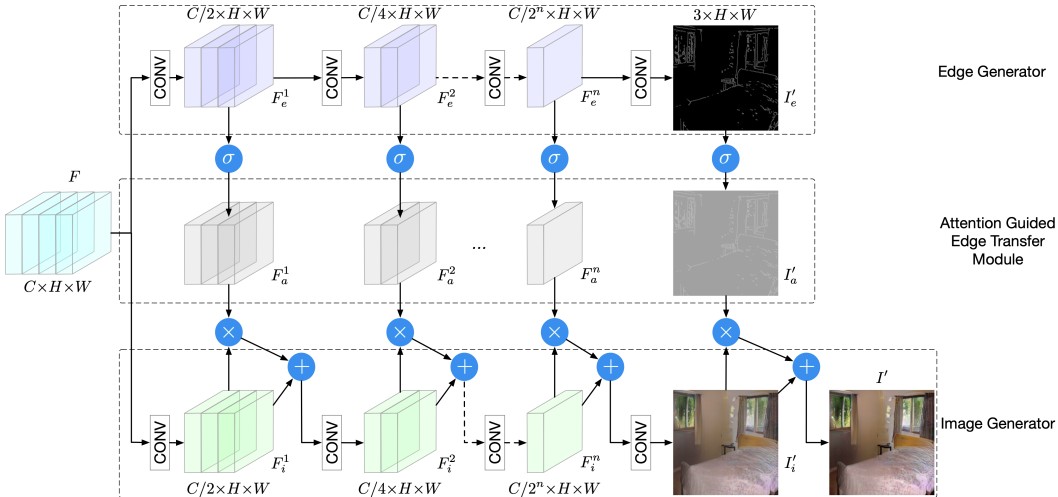

Figure 2: Structure of the proposed edge generator $G_e$, attention guided edge transfer module $G_t$, and image generator $G_i$. $G_e$ selectively transfers useful local structure information to $G_i$ using the proposed attention guided transfer module $G_t$. The symbols $\oplus$, $\otimes$, and $\sigma$ denote element-wise addition, element-wise multiplication, and Sigmoid activation function, respectively.

Meanwhile, to effectively train the network, we propose a multi-modality discriminator $D$ that distinguishes the outputs from both modalities, i.e., edge and image.

### 3.1 EDGE GUIDED SEMANTIC IMAGE SYNTHESIS

**Parameter-Sharing Encoder.** The backbone encoder $E$ can employ any deep network architecture, e.g., the commonly used AlexNet (Krizhevsky et al., 2012), VGG (Simonyan & Zisserman, 2015), and ResNet (He et al., 2016). We directly utilize the feature maps from the last convolutional layer as deep feature representations, i.e., $F=E(S)$, where $E$ represents the encoder; $S \in \mathbb{R}^{N \times H \times W}$ is the input label, with $H$ and $W$ as width and height of the input semantic labels, and $N$ as the total number of semantic classes. Optionally, one can always combine multiple intermediate feature maps to enhance the feature representation. The encoder is shared by the edge generator and the image generator. Then, the gradients from the two generators all contribute to updating the parameters of the encoder. This compact design can potentially enhance the deep representations as the encoder can simultaneously learn structure representations from the edge generation branch and appearance representations from the image generation branch.

**Edge Guided Image Generation.** As discussed, the lack of detailed structure or geometry guidance makes it extremely difficult for the generator to produce realistic local structures and details. To overcome this limitation, we propose to adopt the edge as guidance. A novel edge generator $G_e$ is designed to directly generate the edge maps from the input semantic labels. This also facilitates the shared encoder to learn more local structures of the targeted images. Meanwhile, the image generator $G_i$ aims to generate photo-realistic images from the input labels. In this way, the encoder is boosted to learn the appearance information of the targeted images.

Previous works (Park et al., 2019; Liu et al., 2019; Qi et al., 2018; Chen & Koltun, 2017; Wang et al., 2018) directly use deep networks to generate the target image, which is challenging since the network needs to simultaneously learn appearance and structure information from the input labels. In contrast, the proposed method learns structure and appearance separately via the proposed edge generator and image generator. Moreover, the explicit guidance from the ground truth edge maps can also facilitate the training of the encoder. The framework of both edge and image generators is illustrated in Figure 2. Given the feature maps from the last convolutional layer of the encoder, i.e., $F \in \mathbb{R}^{C \times H \times W}$, where $H$ and $W$ are the width and height of the features, and $C$ is the number of channels, the edge generator produces edge features and edge maps which are further utilized to guide the image generator to generate the intermediate image $I'$. The edge generator $G_e$ contains $n$ convolution layers and correspondingly produces $n$ intermediate feature maps $F_e = \{F_e^j\}_{j=1}^n$. After that, another convolution layer with Tanh non-linear activation is utilized to generate the edge map $I'_e \in \mathbb{R}^{3 \times H \times W}$. Meanwhile, the feature maps $F$ is also fed into the image generator $G_i$ to generate $n$

Figure 3: **Left:** Overview of the proposed semantic preserving module $G_s$, which aims at capturing the semantic information and predicts scaling factors conditioned on the combined feature maps $\mathcal{F}$. These learned factors selectively highlight class-dependent feature maps, which are visualized in different colors. The symbols $\oplus$, $\otimes$, and $\textcircled{\sigma}$ denote element-wise addition, element-wise multiplication, and Sigmoid activation function, respectively. **Right:** Visualization of three different feature channels in $\mathcal{F}'$ on Cityscapes, i.e., road, car, and vegetation.

intermediate feature maps $F_i = \{F_i^j\}_{j=1}^n$. Then another convolution operation with Tanh non-linear activation is adopted to produce the intermediate image $I_i' \in \mathbb{R}^{3 \times H \times W}$. In addition, the intermediate edge feature maps $F_e$ and the edge map $I_e'$ are utilized to guide the generation of the image feature maps $F_i$ and the intermediate image $I'$ via the Attention Guided Edge Transfer as detailed below.

**Attention Guided Edge Transfer.** We further propose a novel attention guided edge transfer module $G_t$ to explicitly employ the edge structure information to refine the intermediate image representations. The architecture of the proposed transfer module $G_t$ is illustrated in Figure 2. To transfer useful structure information from edge feature maps $F_e = \{F_e^j\}_{j=1}^n$ to the image feature maps $F_i = \{F_i^j\}_{j=1}^n$, the edge feature maps are firstly processed by a Sigmoid activation function to generate the corresponding attention maps $F_a = \text{Sigmoid}(F_e) = \{F_a^j\}_{j=1}^n$. The attention aims to provide structural information (which cannot be provided by the input label map) within each semantic class. Then, we multiply the generated attention maps with the corresponding image feature maps to obtain the refined maps, which incorporate local structures and details. Finally, the edge refined features are element-wisely summed with the original image features to produce the final edge refined features, which are further fed to the next convolution layer as $F_i^j = \text{Sigmoid}(F_e^j) \times F_i^j + F_i^j \, (j=1, \cdots, n)$. In this way, the image feature maps also contain the local structure information provided by the edge feature maps. Similarly, to directly employ the structure information from the generated edge map $I_e'$ for image generation, we adopt the attention guided edge transfer module to refine the generated image directly with edge information as

$$I' = \text{Sigmoid}(I_e') \times I_i' + I_i', \tag{1}$$

where $I_a' = \text{Sigmoid}(I_e')$ is the generated attention map. We also visualize the results in Figure 10.

## 3.2 SEMANTIC PRESERVING IMAGE ENHANCEMENT

**Semantic Preserving Module**. Due to the spatial resolution loss caused by convolution, normalization, and down-sampling layers, existing models (Wang et al., 2018; Park et al., 2019; Qi et al., 2018; Chen & Koltun, 2017) cannot fully preserve the semantic information of the input labels as illustrated in Figure 7. For instance, the small "pole" is missing, and the large "fence" is incomplete. To tackle this problem, we propose a novel semantic preserving module, which aims to select class-dependent feature maps and further enhance them through the guidance of the original semantic layout. An overview of the proposed semantic preserving module $G_s$ is shown in Figure 3(left). Specifically, the input of the module denoted as $\mathcal{F}$, is the concatenation of the input label $S$, the generated intermediate edge map $I_e'$ and image $I'$, and the deep feature $F$ produced from the shared encoder $E$. Then, we apply a convolution operation on $\mathcal{F}$ to produce a new feature map $\mathcal{F}_c$ with the number of channels equal to the number of semantic categories, where each channel corresponds to a specific semantic category (a similar conclusion can be found in Fu et al. (2019)). Next, we apply the averaging pooling operation on $\mathcal{F}_c$ to obtain the global information of each class, followed by a Sigmoid activation function to derive scaling factors $\gamma'$ as in $\gamma' = \text{Sigmoid}(\text{AvgPool}(\mathcal{F}_c))$, where each value represents the importance of the corresponding class. Then, the scaling factor $\gamma'$ is adopted to reweight the feature map $\mathcal{F}_c$ and highlight corresponding class-dependent feature maps. The reweighted feature map is further added with the original feature $\mathcal{F}_c$ to compensate for information loss due to multiplication, and produces $\mathcal{F}_c' = \mathcal{F}_c \times \gamma' + \mathcal{F}_c$, where $\mathcal{F}_c' \in \mathbb{R}^{N \times H \times W}$.

After that, we perform another convolution operation on $\mathcal{F}_c'$ to obtain the feature map $\mathcal{F}' \in \mathbb{R}^{(C+N+3+3) \times H \times W}$ to enhance the representative capability of the feature. In addition, $\mathcal{F}'$

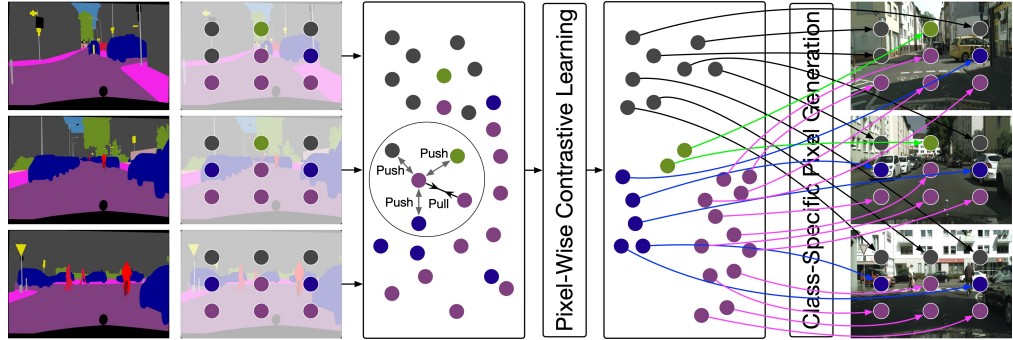

Figure 4: Current semantic image synthesis methods learn to map pixels to an embedding space but ignore intrinsic structures of labeled training data (i.e., inter-layout relations among pixels from the same class, marked with the same color). Our proposed approach, pixel-wise contrastive learning, fosters a new training strategy by explicitly addressing intra-class compactness and inter-class dispersion. By pulling pixels of the same class closer and pushing pixels from different classes apart, our method can create a better-structured embedding space, which leads to the same class generating more similar image content and improves the performance of semantic image synthesis.

has the same size as the original input one $\mathcal{F}$, which makes the module flexible and can be plugged into other existing architectures without modifications of other parts to refine the output. In Figure 3(right), we visualize three channels in $\mathcal{F}'$ on Cityscapes, i.e., road, car, and vegetation. We can easily observe that each channel learns well the class-level deep representations.

Finally, the feature map $\mathcal{F}'$ is fed into a convolution layer followed by a Tanh non-linear activation layer to obtain the final result $I''$. Our semantic preserving module enhances the representational power of the model by adaptively recalibrating semantic class-dependent feature maps and shares similar spirits with style transfer (Huang & Belongie, 2017), and SENet (Hu et al., 2018) and EncNet (Zhang et al., 2018a). One intuitive example of the utility of the module is for the generation of small object classes: these classes are easily missed in the generation results due to spatial resolution loss, while our scaling factor can put an emphasis on small objects and help preserve them.

**Similarity Loss.** Preserving semantic information from isolated pixels is very challenging for deep networks. To explicitly enforce the network to capture the relationship between semantic categories, a new similarity loss is introduced. This loss forces the network to consider both intra-class and inter-class pixels for each pixel in the label. Specifically, a state-of-the-art pretrained model (i.e., Seg-Former (Xie et al., 2021)) is used to transfer the generated image $I''$ back to a label $S'' \in \mathbb{R}^{N \times H \times W}$, where $N$ is the total number of semantic classes, and $H$ and $W$ represent the width and height of the image, respectively. A conventional method uses the cross entropy loss between $S''$ and $S$ to address this problem. However, such a loss only considers the isolated pixel while ignoring the semantic correlation with other pixels.

To address this limitation, we construct a similarity map from $S \in \mathbb{R}^{N \times H \times W}$. Firstly, we reshape $S$ to $\hat{S} \in \mathbb{R}^{N \times M}$, where $M = HW$. Next, we perform a matrix multiplication to obtain a similarity map $A = \hat{S}\hat{S}^{\top} \in \mathbb{R}^{M \times M}$. This similarity map encodes which pixels belong to the same category, meaning that if the j-*th* pixel and the i-*th* pixel belong to the same category, then the value of the j-*th* row and the i-*th* column in $A$ is 1; otherwise, it is 0. Similarly, we can obtain a similarity map $A''$ from the label $S''$. Finally, we calculate the binary cross entropy loss between the two similarity maps $\{a_m \in A, m \in [1, M^2]\}$ and $\{a''_m \in A'', m \in [1, M^2]\}$ as

$$\mathcal{L}_{sim}(S, S'') = -\frac{1}{M^2} \sum_{m=1}^{M^2} (a_m \log a''_m + (1 - a_m) \log(1 - a''_m)). \quad (2)$$

This similarity loss explicitly captures intra-class and inter-class semantic correlation, leading to better generation results.

### 3.3 CONTRASTIVE LEARNING FOR SEMANTIC IMAGE SYNTHESIS

**Pixel-Wise Contrastive Learning.** Existing semantic image synthesis models use deep networks to map labeled pixels to a non-linear embedding space. However, these models often only take

into account the "local" context of pixel samples within an individual input semantic layout, and fail to consider the "global" context of the entire dataset, which includes the semantic relationships between pixels across different input layouts. This oversight raises an important question: what should the ideal semantic image synthesis embedding space look like? Ideally, such a space should not only enable accurate categorization of individual pixel embeddings, but also exhibit a well-structured organization that promotes intra-class similarity and inter-class difference. That is, pixels from the same class should generate more similar image content than those from different classes in the embedding space. Previous approaches to representation learning propose that incorporating the inherent structure of training data can enhance feature discriminativeness. Hence, we conjecture that despite the impressive performance of existing algorithms, there is potential to create a more well-structured pixel embedding space by integrating both the local and global context.

The objective of unsupervised representation learning is to train an encoder that maps each training semantic layout $S$ to a feature vector $v = B(S)$, where $B$ represents the backbone encoder network. The resulting vector $v$ should be an accurate representation of $S$. To accomplish this task, contrastive learning approaches use a training method that distinguishes a positive from multiple negatives, based on the similarity principle between samples. The InfoNCE (Oord et al., 2018; Gutmann & Hyvärinen, 2010) loss function, a popular choice for contrastive learning, can be expressed as

$$\mathcal{L}_S = -\log \frac{\exp(v \cdot v_+/\tau)}{\exp(v \cdot v_+/\tau) + \sum_{v_- \in N_S} \exp(v \cdot v_-/\tau)}, \tag{3}$$

where $v_+$ represents an embedding of a positive for $S$, and $N_S$ includes embeddings of negatives. The symbol "$\cdot$" refers to the inner (dot) product, and $\tau > 0$ is a temperature hyper-parameter. It is worth noting that the embeddings used in the loss function are normalized using the $L_2$ method.

One limitation of this training objective design is that it only penalizes pixel-wise predictions independently, without considering the cross-relationship between pixels. To overcome this limitation, we take inspiration from (Wang et al., 2021; Khosla et al., 2020) and propose a contrastive learning method that operates at the pixel level and is intended to regularize the embedding space while also investigating the global structures present in the training data (see Figure 4). Specifically, our contrastive loss computation uses training semantic layout pixels as data samples. For a given pixel $i$ with its ground-truth semantic label $c$, the positive samples consist of other pixels that belong to the same class $c$, while the negative samples include pixels belonging to other classes $C \backslash c$. As a result, the proposed pixel-wise contrastive learning loss is defined as follows

$$\mathcal{L}_i = \frac{1}{|P_i|} \sum_{i_+ \in P_i} -\log \frac{\exp(i \cdot i_+/\tau)}{\exp(i \cdot i_+/\tau) + \sum_{i_- \in N_i} \exp(i \cdot i_-/\tau)}. \tag{4}$$

For each pixel $i$, we use $P_i$ and $N_i$ to represent the pixel embedding collections of positive and negative samples, respectively. Importantly, the positive and negative samples and the anchor $i$ are not required to come from the same layout. The goal of this pixel-wise contrastive learning approach is to create an embedding space in which same-class pixels are pulled closer together, and different-class pixels are pushed further apart. The result of this process is that pixels with the same class generate image contents that are more similar, which can lead to superior generation performance.

**Class-Specific Pixel Generation.** To overcome the challenges posed by training data imbalance between different classes and size discrepancies between different semantic objects, we introduce a new approach that is specifically designed to generate small object classes and fine details. Our proposed method is a class-specific pixel generation approach that focuses on generating image content for each semantic class. By doing so, we can avoid the interference from large object classes during joint optimization, and each subgeneration branch can concentrate on a specific class generation, which in turn results in similar generation quality for different classes and yields richer local image details.

An overview of the class-specific pixel generation method is provided in Figure 4. After the proposed pixel-wise contrastive learning, we obtain a class-specific feature map for each pixel. Then, the feature map is fed into a decoder for the corresponding semantic class, which generates an output image $\hat{I}_i$. Since we have the proposed contrastive learning loss, we can use the parameter-shared decoder to generate all classes. To better learn each class, we also utilize a pixel-wise $L_1$ reconstruction loss, which can be expressed as $\mathcal{L}_{L_1} = \sum_{i=1}^{N} \mathbb{E}_{I_i, \hat{I}_i}[||I_i - \hat{I}_i||_1]$. The final output $I_g$ from

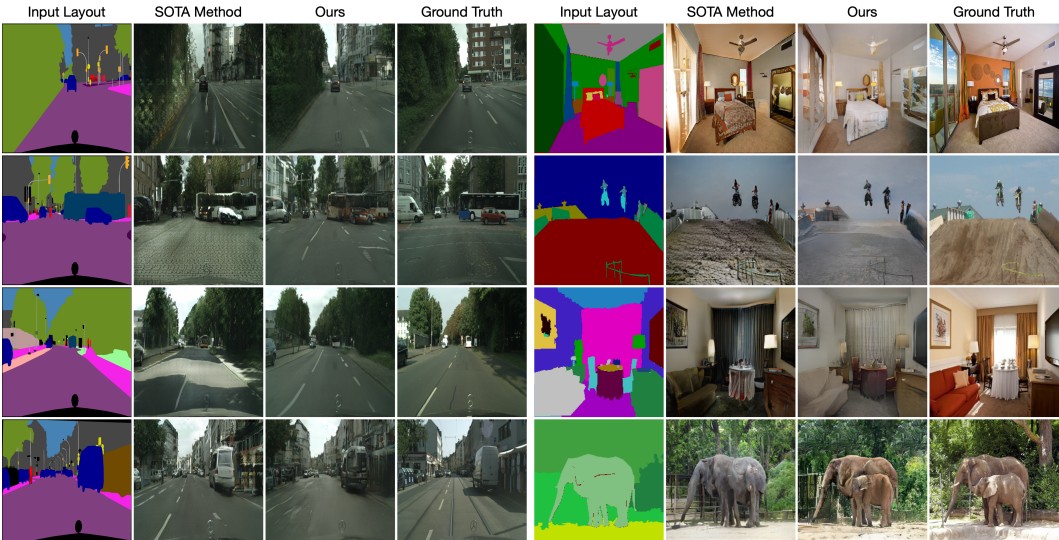

Figure 5: Existing SOTA method (i.e., OASIS) vs. our proposed ECGAN on three datasets. Cityscapes: left; ADE20K: top right two; COCO-Stuff: bottom right two.

Table 1: User study on Cityscapes, ADE20K, and COCO-Stuff. The numbers indicate the percentage of users who favor the results of the proposed method over the competing methods.

| AMT ↑ | Cityscapes | ADE20K | COCO-Stuff |
|---|---|---|---|
| Ours vs. CRN (Chen & Koltun, 2017) | 88.8 ± 3.4 | 94.8 ± 2.7 | 95.3 ± 2.1 |
| Ours vs. Pix2pixHD (Wang et al., 2018) | 87.2 ± 2.9 | 93.6 ± 3.1 | 93.9 ± 2.4 |
| Ours vs. SIMS (Qi et al., 2018) | 85.3 ± 3.8 | - | - |
| Ours vs. GauGAN (Park et al., 2019) | 84.7 ± 4.3 | 88.4 ± 3.7 | 90.8 ± 2.5 |
| Ours vs. DAGAN (Tang et al., 2020a) | 81.8 ± 3.9 | 86.2 ± 3.6 | - |
| Ours vs. CC-FPSE (Liu et al., 2019) | 79.5 ± 4.2 | 85.1 ± 3.9 | 86.7 ± 2.8 |
| Ours vs. LGGAN (Tang et al., 2020b) | 78.4 ± 4.7 | 82.7 ± 4.5 | - |
| Ours vs. OASIS (Sushko et al., 2021) | 76.7 ± 4.8 | 80.6 ± 4.5 | 82.5 ± 3.1 |

the pixel generation network can be obtained by performing an element-wise addition of all the class-specific outputs: $I_g = I_{g_1} \oplus I_{g_2} \oplus \cdots \oplus I_{g_N}$.

## 4 EXPERIMENTS

**Datasets and Evaluation Metrics.** We follow GauGAN (Park et al., 2019) and conduct experiments on three datasets, i.e., Cityscapes (Cordts et al., 2016), ADE20K (Zhou et al., 2017), and COCO-Stuff (Caesar et al., 2018). We employ the mean Intersection-over-Union (mIoU), Pixel Accuracy (Acc), and Fréchet Inception Distance (FID) (Heusel et al., 2017) as the evaluation metrics.

**State-of-the-Art Comparisons.** We adopt GauGAN as the encoder $E$ to validate the effectiveness of the proposed method. Visual comparison results on all three datasets with the SOTA method are shown in Figure 5. ECGAN achieves visually better results with fewer visual artifacts than the SOTA method. For instance, on scene datasets such as Cityscapes and ADE20K, ECGAN generates sharper images than the baseline, especially at local structures and details.

Moreover, we follow the same evaluation protocol as GauGAN and conduct a user study. Specifically, we provide the participants with an input layout and two generated images from different models and ask them to choose the generated image that looks more like a corresponding image of the layout. The users are given unlimited time to make the decision. For each comparison, we randomly generate 400 questions for each dataset, and each question is answered by 10 different participants. For other methods, we use the public code and pretrained models provided by the authors to generate images. As shown in Table 1, users favor our synthesized results on all three datasets compared with other competing methods, further validating that the generated images by ECGAN are more natural. Although the user study is more suitable for evaluating the quality of the generated images, we also follow previous works and use mIoU, Acc, and FID for quantitative evaluation. The

Table 2: Quantitative comparison of different methods on Cityscapes, ADE20K, and COCO-Stuff.

| Method | Cityscapes | | | ADE20K | | | COCO-Stuff | | |
|---|---|---|---|---|---|---|---|---|---|
| | mIoU ↑ | Acc ↑ | FID ↓ | mIoU ↑ | Acc ↑ | FID ↓ | mIoU ↑ | Acc ↑ | FID ↓ |
| CRN (Chen & Koltun, 2017) | 52.4 | 77.1 | 104.7 | 22.4 | 68.8 | 73.3 | 23.7 | 40.4 | 70.4 |
| SIMS (Qi et al., 2018) | 47.2 | 75.5 | 49.7 | - | - | - | - | - | - |
| Pix2pixHD (Wang et al., 2018) | 58.3 | 81.4 | 95.0 | 20.3 | 69.2 | 81.8 | 14.6 | 45.8 | 111.5 |
| GauGAN (Park et al., 2019) | 62.3 | 81.9 | 71.8 | 38.5 | 79.9 | 33.9 | 37.4 | 67.9 | 22.6 |
| DPGAN (Tang & Sebe, 2021) | 65.2 | 82.6 | 53.0 | 39.2 | 80.4 | 31.7 | - | - | - |
| DAGAN (Tang et al., 2020a) | 66.1 | 82.6 | 60.3 | 40.5 | 81.6 | 31.9 | - | - | - |
| SelectionGAN (Tang et al., 2019) | 83.8 | 82.4 | 65.2 | 40.1 | 81.2 | 33.1 | - | - | - |
| SelectionGAN++ (Tang et al., 2022b) | 64.5 | 82.7 | 63.4 | 41.7 | 81.5 | 32.2 | - | - | - |
| LGGAN (Tang et al., 2020b) | 68.4 | 83.0 | 57.7 | 41.6 | 81.8 | 31.6 | - | - | - |
| LGGAN++ (Tang et al., 2022a) | 67.7 | 82.9 | 48.1 | 41.4 | 81.5 | 30.5 | - | - | - |
| CC-FPSE (Liu et al., 2019) | 65.5 | 82.3 | 54.3 | 43.7 | 82.9 | 31.7 | 41.6 | 70.7 | 19.2 |
| OASIS (Sushko et al., 2021) | 69.3 | - | 47.7 | 48.8 | - | 28.3 | 44.1 | - | 17.0 |
| RESAIL (Shi et al., 2022) | 69.7 | **83.2** | 45.5 | 49.3 | 84.8 | 30.2 | 44.7 | 73.1 | 18.3 |
| SAFM (Lv et al., 2022) | 70.4 | 83.1 | 49.5 | 50.1 | **86.6** | 32.8 | 43.3 | **73.4** | 24.6 |
| ECGAN (Ours) | **72.2** | 83.1 | **44.5** | **50.6** | 83.1 | **25.8** | **46.3** | 70.5 | **15.7** |

Table 3: Ablation study of the proposed ECGAN on Cityscapes, ADE20K, and COCO-Stuff.

| # | Setting | Cityscapes | | | ADE20K | | | COCO-Stuff | | |
|---|---|---|---|---|---|---|---|---|---|---|
| | | mIoU ↑ | Acc ↑ | FID ↓ | mIoU ↑ | Acc ↑ | FID ↓ | mIoU ↑ | Acc ↑ | FID ↓ |
| B1 | $E+G_i$ | 58.6 | 81.4 | 65.7 | 36.9 | 78.5 | 38.2 | 36.8 | 65.1 | 24.5 |
| B2 | $E+G_i+G_e$ | 60.2 | 81.7 | 61.0 | 38.7 | 79.2 | 36.3 | 37.5 | 66.3 | 22.9 |
| B3 | $E+G_i+G_e+G_t$ | 61.5 | 82.0 | 59.0 | 40.6 | 80.3 | 34.6 | 39.1 | 67.0 | 21.7 |
| B4 | $E+G_i+G_e+G_t+G_s$ | 64.5 | 82.5 | 57.1 | 42.0 | 82.0 | 32.4 | 41.4 | 68.2 | 19.8 |
| B5 | $E+G_i+G_e+G_t+G_s+G_l$ | 66.8 | 82.7 | 52.2 | 45.8 | 82.4 | 29.9 | 43.7 | 69.1 | 17.6 |
| B6 | $E+G_i+G_e+G_t+G_s+G_l+G_c$ | **72.2** | **83.1** | **44.5** | **50.6** | **83.1** | **25.8** | **46.3** | **70.5** | **15.7** |

results of the three datasets are shown in Table 2. The proposed ECGAN outperforms other leading methods by a large margin on all three datasets, validating the effectiveness of the proposed method.

**Ablation Study.** We conduct extensive ablation studies on three datasets to evaluate different components of the proposed ECGAN. Our method has six baselines (i.e., B1, B2, B3, B4, B5, B6) as shown in Table 3: B1 means only using the encoder $E$ and the proposed image generator $G_i$ to synthesize the targeted images. B2 means adopting the proposed image generator $G_i$ and edge generator $G_e$ to simultaneously produce both edge maps and images. B3 connects the image generator $G_i$ and the edge generator $G_e$ by using the proposed attention guided edge transfer module $G_t$. B4 employs the proposed semantic preserving module $G_s$ to further improve the quality of the final results. B5 uses the proposed label generator $G_l$ to produce the label from the generated image and then calculate the similarity loss between the generated label and the real one. B6 is our full model and uses the proposed pixel-wise contrastive learning and class-specific pixel generation methods to capture more semantic relations by explicitly exploring the structures of labeled pixels from multiple input semantic layouts. As shown in Table 3, each proposed module improves the performance on all three metrics, validating the effectiveness. We also provide more ablation analysis in Appendix.

## 5 CONCLUSION

We propose a novel ECGAN for semantic image synthesis. It introduces four core components: edge guided image generation strategy, attention guided edge transfer module, semantic preserving module, and contrastive learning module. The first one is employed to generate edge maps from input semantic labels. The second one is used to selectively transfer the useful structure information from the edge branch to the image branch. The third one is adopted to alleviate the problem of spatial resolution losses caused by different operations in the deep nets. The last one is utilized to investigate global semantic relations between training pixels, guiding pixel embeddings towards cross-image category-discriminative representations. Extensive experiments on three datasets show that ECGAN achieves significantly better results than existing models.

## 6 ACKNOWLEDGMENT

This work was partly supported by ETH General Fund, the Alexander von Humboldt Foundation, and the EU H2020 project AI4Media under Grant 951911.

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

Appendix

## A MODEL TRAINING

**Multi-Modality Discriminator.** To facilitate the training of the proposed method for high-quality edge and image generation, a novel multi-modality discriminator is developed to simultaneously distinguish outputs from two modality spaces, i.e., edge and image. Since the edges and RGB images share the same structure, they can be learned using the multi-modality discriminator. In the preliminary experiment, we also tried to use two discriminators (i.e., an edge discriminator and an image discriminator), but no performance improvement was observed while increasing the model complexity. Thus, we use the proposed multi-modality discriminator. The framework of the multi-modality discriminator is shown in Figure 1, which is capable of discriminating both real/fake images and edges. To discriminate real/fake edges, the discriminator loss considering the semantic label $S$ and the generated edge $I'_e$ (or the real edge $I_e$) is as

$$\mathcal{L}_{\text{CGAN}}(G_e, D) = \mathbb{E}_{S, I_e} \left[\log D(S, I_e)\right] + \mathbb{E}_{S, I'_e} \left[\log(1 - D(S, I'_e))\right], \tag{5}$$

which guides the model to distinguish real edges from fake generated edges. Further, to discriminate real/fake images, the discriminator loss regarding semantic label $S$ and the generated images $I'$, $I''$ (or the real image $I$) is as Eq. 6, which guides the model to discriminate real/fake images,

$$\mathcal{L}_{\text{CGAN}}(G_i, G_s, D) = (\lambda + 1)\mathbb{E}_{S, I} \left[\log D(S, I)\right] + \mathbb{E}_{S, I'} \left[\log(1 - D(S, I'))\right] \\ + \lambda \mathbb{E}_{S, I''} \left[\log(1 - D(S, I''))\right], \tag{6}$$

where $\lambda$ controls the losses of the two generated images. The inclusion of $I'$ and $I''$ is a cascaded coarse-to-fine generation strategy (Tang et al., 2019), i.e., $I'$ is the coarse result, while $I''$ is the refined result. The intuition is that $I''$ will be better generated based on $I'$, so we provide $I'$ to the discriminator to ensure that $I'$ is also realistic.

**Optimization Objective.** Equipped with the multi-modality discriminator, we elaborate on the training objective for the proposed method as follows. Five different losses, i.e., the multi-modality adversarial loss, the similarity loss, the contrastive learning loss, the discriminator feature matching loss $\mathcal{L}_f$, and the perceptual loss $\mathcal{L}_p$ are used to optimize the proposed ECGAN,

$$\min_G \max_D \mathcal{L} = \lambda_c \underbrace{(\mathcal{L}_{\text{CGAN}}(G_e, D) + \mathcal{L}_{\text{CGAN}}(G_i, G_s, D))}_{\text{Multi-Modality Adversarial Loss}} + \lambda_s \underbrace{\mathcal{L}_{sim}(S, S') + \mathcal{L}_{sim}(S, S'')}_{\text{Similarity Loss}}$$
$$+ \lambda_l \underbrace{\mathcal{L}_i + \mathcal{L}_{L_1}}_{\text{Contrastive Learning Loss}} + \lambda_f \underbrace{(\mathcal{L}_f(I_e, I'_e) + \mathcal{L}_f(I, I') + \lambda \mathcal{L}_f(I, I''))}_{\text{Discriminator Feature Matching Loss}} \tag{7}$$
$$+ \lambda_p \underbrace{(\mathcal{L}_p(I_e, I'_e) + \mathcal{L}_p(I, I') + \lambda \mathcal{L}_p(I, I''))}_{\text{Perceptual Loss}},$$

where $\lambda_c$, $\lambda_s$, $\lambda_l$, $\lambda_f$, and $\lambda_p$ are the parameters of the corresponding loss that contributes to the total loss $\mathcal{L}$; where $\mathcal{L}_f$ matches the discriminator intermediate features between the generated images/edges and the real images/edges; where $\mathcal{L}_p$ matches the VGG extracted features between the generated images/edges and the real images/edges. By maximizing the discriminator loss, the generator is promoted to simultaneously generate reasonable edge maps that can capture the local-aware structure information and generate realistic images semantically aligned with the input labels.

## B IMPLEMENTATION DETAILS

For both the image generator $G_i$ and edge generator $G_e$, the kernel size and padding size of convolution layers are all $3 \times 3$ and 1 for preserving the feature map size. We set $n = 3$ for generators $G_i$, $G_s$, and $G_t$. The channel size of feature $F$ is set to $C = 64$. For the semantic preserving module $G_s$, we adopt an adaptive average pooling operation. Spectral normalization (Miyato et al., 2018) is applied to all the layers in both the generator and discriminator. We adopt the Canny edge detector (Canny, 1986) to extract edge maps for training. During the testing phase, we do not need to use additional data. Thus the comparison with existing methods is fair. When calculating the contrastive learning loss, we found that the more layouts are used, the better the performance is. When more

than 8 layouts are used as input, the performance improvement is not obvious, but the training of the whole model will become slow. Therefore, considering the balance between performance and time, we finally choose 8 layouts as input to calculate the contrastive learning loss.

Also, we follow the training procedures of GANs (Goodfellow et al., 2014) and alternatively train the generator $G$ and discriminator $D$, i.e., one gradient descent step on the discriminator and generator alternately. We use the Adam solver (Kingma & Ba, 2015) and set $\beta_1{=}0$, $\beta_2{=}0.999$. $\lambda_c$, $\lambda_s$, $\lambda_l$, $\lambda_f$, and $\lambda_p$ in Eq. 7 is set to 1, 1, 1, 10, and 10, respectively. All $\lambda$ in both Eq. 6 and 7 are set to 2. We conduct the experiments on an NVIDIA DGX1 with 8 V100 GPUs.

## C    ADDITIONAL RESULTS

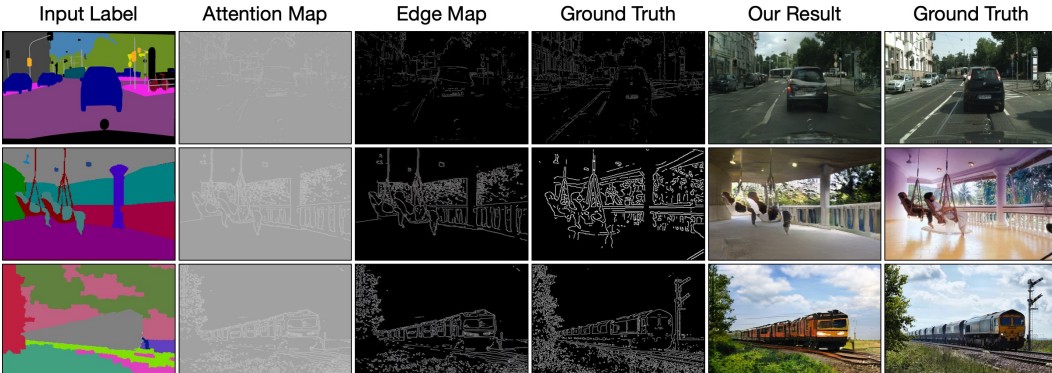

Figure 6: Edge and attention maps generated by the proposed method.

**Visualization of Edge and Attention Maps.** We also visualize the generated edge and attention maps in Figure 6. We observe that the proposed method can generate reasonable edge maps according to the input labels. Thus the generated edge maps can be used to provide more local structure information for generating more photo-realistic images.

**Visualization of Segmentation Maps.** We follow GauGAN and apply pre-trained segmentation networks (Yu et al., 2017; Xiao et al., 2018) on the generated images to produce segmentation maps. Results compared with the baseline method are shown in Figure 7. We observe that the proposed method consistently generates better semantic labels than the baseline on both datasets.

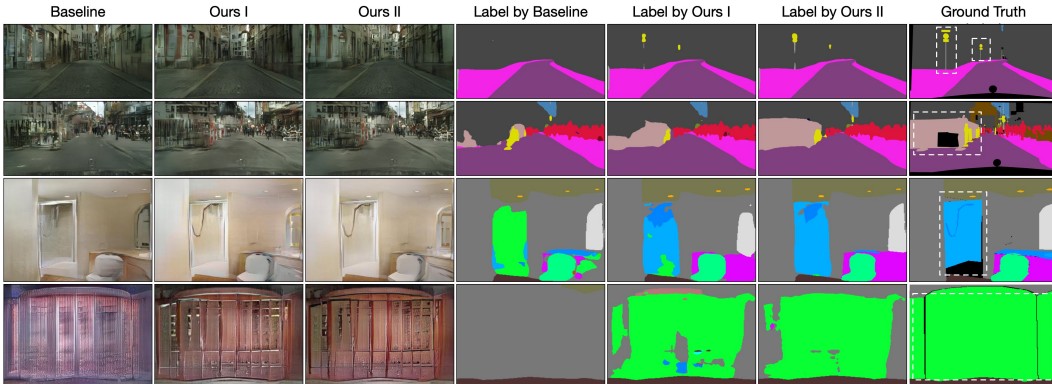

Figure 7: Segmentation labels generated by the baseline and our proposed method. "Ours I" and "Ours II" stand for $I'$ and $I''$, respectively.

**Multi-Modal Image and Edge Synthesis.** We follow GauGAN (Park et al., 2019) and apply a style encoder and a KL-Divergence loss with a loss weight of 0.05 to enable multi-modal image and edge synthesis. As shown in Figure 8, our model generates different edges and images from the same input layout, which we believe will benefit other tasks, e.g., image inpainting and super-resolution.

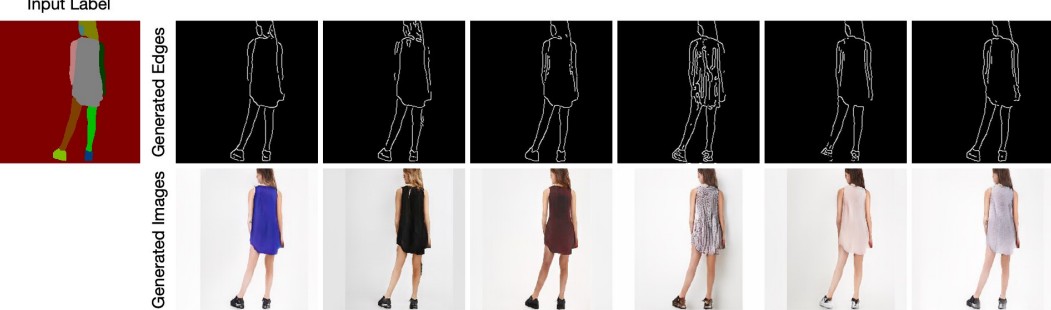

Figure 8: Results generated by the proposed method for multi-modal image and edge synthesis.

Table 4: Multi-modal synthesis evaluation on ADE20K.

| Method | Multi-Modal | LPIPS ↑ |
|---|---|---|
| GauGAN+ (Park et al., 2019) | Encoder | 0.16 |
| GauGAN+ (Park et al., 2019) | 3D Noise | 0.50 |
| OASIS (Sushko et al., 2021) | 3D Noise | 0.35 |
| ECGAN (Ours) | Encoder | 0.18 |
| ECGAN (Ours) | 3D Noise | **0.52** |

Table 5: mIoU of small objects on Cityscapes.

| mIoU ↑ | Pole | Light | Sign | Rider | Mbike | Bike | Overall |
|---|---|---|---|---|---|---|---|
| OASIS (Sushko et al., 2021) | 23.4 | 32.6 | 14.9 | 27.3 | 31.2 | 26.6 | 26.0 |
| ECGAN (Ours) | **26.2** | **36.7** | **17.4** | **30.2** | **33.5** | **28.7** | **28.8** |

Moreover, we follow OASIS (Sushko et al., 2021) and use LPIPS (Zhang et al., 2018b) to evaluate the variation in the multi-model image synthesis on the ADE20K dataset. Following in OASIS, we generate 20 images and compute the mean pairwise scores, and then average over all label maps. The higher the LPIPS scores, the more diverse the generated images are. We follow OASIS and GauGAN, and employ two settings (i.e., encoder and 3D noise) to evaluate multi-modal image synthesis. Table 4 shows that the proposed ECGAN achieves better results than OASIS and GauGAN in both settings. Note that existing methods (e.g., OASIS (Sushko et al., 2021) and GauGAN (Park et al., 2019)) can only achieve multi-modal image synthesis.

**Evaluation Focused on Small Objects.** We report mIoU on six small object categories of Cityscapes (i.e., pole, light, sign, rider, mbike, and bike) in Table 5. Our ECGAN generates better mIoU than the SOTA method (i.e., OASIS (Sushko et al., 2021)) on all these small object classes. We also show visualization results in Figure 9, clearly confirming that the proposed method is highly capable of preserving small objects in the output.

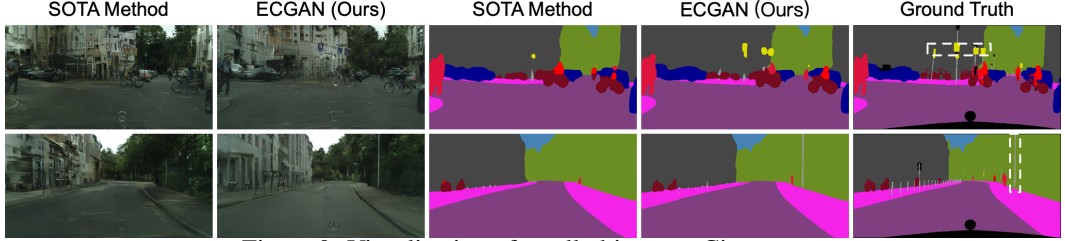

Figure 9: Visualization of small objects on Cityscapes.

# D   ADDITIONAL ABLATION STUDY

The ablation study results are shown in Table 3.

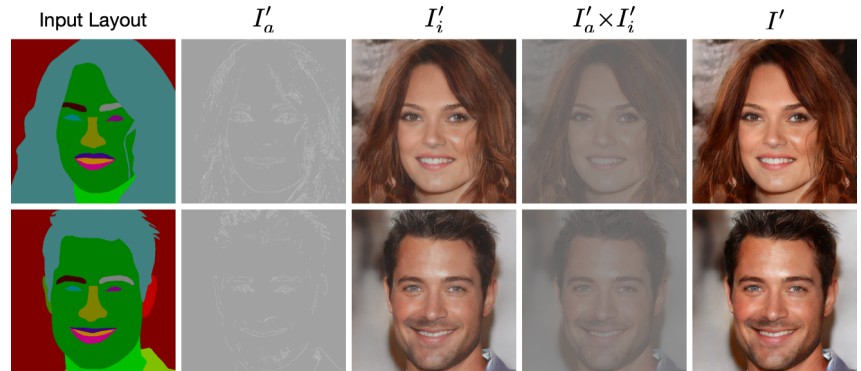

Figure 10: Visualization of the differences after the edge-guided refinement in Eq. 1.

**Effect of Edge Guided Generation Strategy.** When using the edge generator $G_e$ to produce the corresponding edge map from the input label, performance on all evaluation metrics is improved. We also provide several visualization results of the differences (see Eq. 1) after the edge-guided refinement in Figure 10.

**Effect of Attention Guided Edge Transfer Module.** We observe that the implicitly learned edge structure information by the "$E+G_i+G_e$" baseline is not enough for such a challenging task. Thus we further adopt the transfer module $G_t$ to transfer useful edge structure information from the edge generation branch to the image generation branch. We observe that performance gains are obtained on the mIoU, Acc, and FID metrics in all three datasets. This means that $G_t$ indeed learns rich feature representations with more convincing structure cues and details and then transfers them from the generator $G_e$ to the generator $G_i$.

**Effect of Semantic Preserving Module.** By adding $G_s$, the overall performance is further boosted on all the three datasets. This means $G_s$ indeed learns and highlights class-specific semantic feature maps, leading to better generation results. In Figure 7, we show some samples of the generated semantic maps. We observe that the semantic maps produced by the results with $G_s$ (i.e., "Label by Ours II" in Figure 7) are more accurate than those without using $G_s$ ("Label by Ours I" in Figure 7). Moreover, we visualize three channels in $\mathcal{F}'$ on Cityscapes in Figure 3(right), i.e., road, car, and vegetation. Each channel learns well the class-level deep representations.

**Effect of Similarity Loss.** By adding the proposed label generator $G_l$ and similarity loss, the overall performance is further boosted on all three metrics. This means the proposed similarity loss indeed captures more intra-class and inter-class semantic dependencies, leading to better semantic layouts in the generated images.

**Effect of Contrastive Learning.** When adopting the proposed pixel-wise contrastive learning and class-specific pixel generation methods to produce the results, the performance is significantly improved on all three datasets on all three evaluation metrics. This means that the model does indeed learn a more discriminative class-specific feature representation, confirming the superiority of our design.

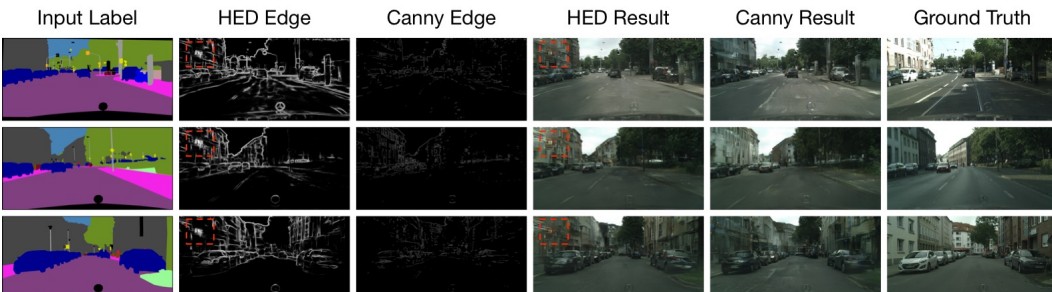

Figure 11: HED vs. Canny edge extraction.

**Effect of Edge Extraction Methods.** We also conduct experiments on Cityscapes with HED (Xie & Tu, 2015), leading to the following results: 56.7 (FID), 64.5 (mIoU), and 82.3 (Acc), which are slightly worse than the results of Canny in Table 3. The reason is that the edges from HED are very thick and cannot accurately represent the edge of objects. It also ignores some local details since it focuses on extracting the contours of objects. Thus, HED is unsuitable for our setting as we aim to generate more local details/structures. Moreover, we see that the generated HED edges contain artifacts, as indicated in the red boxes in Figure 11, which makes the generated images tend to have blurred edges.

