# OpenReview forum: "Edge Guided GANs with Contrastive Learning for Semantic Image Synthesis"
_ICLR.cc/2023/Conference — ICLR 2023 poster_

### Official Review · Reviewer_3rGQ · 2022-10-22

**Confidence:** 5
**Correctness:** 2
**Technical Novelty And Significance:** 2
**Empirical Novelty And Significance:** 2
**Recommendation:** 6

**Clarity, Quality, Novelty And Reproducibility:**

Although the writing is good, the clarity of proposed method is hard to follow, especially, it's not clear how and where the contrastive learning works, how to constrain the Edge Generator and Image Generator to produce edges and images respectively, and why the Semantic Preserving Module can perform to preserve semantics considering the existence of $G_l$. Thus, the reproducibility is also not good. As mentioned in the weaknesses, the novelty is limited. Overall, the quality of this work is fair.

**Strength And Weaknesses:**

Strenghts:
  - The writing is good.
  - The raised three issues make some sense for semantic image synthesis.

Weaknesses:
  - The design is complicated and the novelty is limited. The proposed method, namely ECGAN, contains a lot of CNN-based modules with different given functions, for example, Edge Generator for producing edge maps from semantic maps, Image Generator for generating intermediate images from semantic maps, as well as Semantic Preserving Module and Multi-Modality Discriminator, etc., but actually, it's not sure or clear how these modules perform as the given definitions. More importantly, for the three solutions dealing with the three issues, the contributions are limited, for instance, edge guided semantic image synthesis is not new ("Edge Guided GANs with Semantic Preserving for Semantic Image Synthesis", arXiv:2003.13898), and the similar idea on pixel-wise contrastive learning has been explored on semantic segmentation ("Contrastive Learning for Label Efficient Semantic Segmentation", ICCV 2021; "Region-Aware Contrastive Learning for Semantic Segmentation", ICCV 2021).
  - The evaluation is not sufficient without considering the state-of-the-art methods. The state-of-the-art methods about semantic image synthesis include the works published on the recent CVPR 2022, like "Retrieval-based Spatially Adaptive Normalization for Semantic Image Synthesis", "Semantic-shape Adaptive Feature Modulation for Semantic Image Synthesis", as well as "Alleviating Semantics Distortion in Unsupervised Low-Level Image-to-Image Translation via Structure Consistency Constraint". Moreover, the ablation study in the Appendix is somehow insufficient to validate the efficacy of the design, since it's better to show that each component indeed performs as given definitions.

**Summary Of The Paper:**

To address semantic image synthesis, this paper mainly proposes to tackle three issues of lack of details from semantic lables, spatial resolution loss from CNN operations, and ignoring 'global' semantic information from a single input semantic layout, with the design of edge guided generative adversarial network (GAN), semantic perserving module, and pixel-wise contrastive learning, respectively. This work seems complicated combining some existing techniques with some new changes.

**Summary Of The Review:**

Considering the weaknesses of limited novelty and insufficient evaluation, as well as the issues of clarity and reproducibility, I lean towards rejecting.

------
After discussion with AC and the other reviewers, I agree to marginally accept the paper.

---

### Official Review · Reviewer_QAjR · 2022-10-24

**Confidence:** 4
**Correctness:** 4
**Technical Novelty And Significance:** 2
**Empirical Novelty And Significance:** 2
**Recommendation:** 6

**Clarity, Quality, Novelty And Reproducibility:**


# Clarity

* Some important material is not described in the main paper, and deferred to the supplementary. For example, the loss used to train the model (Appendix A on page 13) and ablation studies that consider the impact of the different model components (Appendix D on page 15).
The last paragraph of section 3.2 on Class-specific pixel generation was unclear to me.

* Figure 5 does not state what state of the art model was used for comparison, it is also not indicated to which datasets the images correspond.

* The introduction on page 2 discusses a hypothetical case of using a separate synthesis network for each semantic class, and suggests that in such a case the number of parameters would grow exponentially in the number of classes. The latter claim seems unfounded, or is at least unclear. If the authors address this (claimed) shortcoming of existing work I'd expect a quantitative analysis of this.



# Quality
* The experiments follow the standard protocol for the task, including the considered datasets and metrics.

* On several occasions the authors discuss the difficulty of existing methods to generate small objects (eg abstract, and related work). The experimental work, however, does not specifically analyse the generation ability of existing and the proposed models for such objects.

# Novelty
* The paper offers several new ideas for semantic image synthesis, including the edge-driven generation mechanism, contrastive feature learning, and semantic similarity loss.

# Reproducibility
* The code of the described method is not released.
* The user study is described in very little detail, and would be hard to reproduce. How many images were used, how many annotators, how were the images of various models obtained, no code nor images are released by the authors, etc.
* Experiments are conducted on a fairly standard DGX1 hardware platform with eight V100 GPUs.

**Strength And Weaknesses:**


# Strengths

* Edge information is used as a guiding signal to improve semantic image synthesis

* Contrastive learning is used to improve feature learning in the semantic synthesis pipeline

* Experimental results improve over a collection of strong baseline methods.


# Weaknesses

* Not all components of the method are clearly explained in the paper. In particular  the class-specific pixel generation approach was unclear.

* The paper is lacking detail regarding some technical material and experiments, see below.

* The paper is unclear in how it introduces diversity among the generated images, and results are not analysed quantitatively or qualitatively for diversity. It would  strengthen the paper to clarify this and include such an evaluation, eg using the LPIPS metric following OASIS.

**Summary Of The Paper:**

This paper proposes a novel semantic image synthesis model (semantic segmentation map in, RGB image out) based on adversarial training.
The main idea is to use edge maps as a structure to guide the image synthesis process.
To this end a special edge-generating branch is used to generate an edge map from the semantic input map.
The image generating branch leverages the generated edge map (and intermediate features) via an ``attention'' mechanism to update the image synthesis features.
Training is based on conditional discriminators for edge maps and RGB images (canny edge maps are used for ground truth).
Additionally, a binary similarity loss is used to encourages pixels that should belong to the same class according to the input maps to take the same label according a pre-trained segmentation network.
Finally, pixel-wise contrastive learning is employed to improve results, which consists in encouraging pixels of the same class to have similar features.



**Summary Of The Review:**


The paper introduces some novel ideas, and shows quantitative evaluation results improving over state-of-the-art methods, as well as user preference in a user study.
Some parts of the paper lack clarity, and analysis of generation diversity is not included.

### Post-rebuttal comment ###
Based on the author rebuttal and other reviews, I'm upgrading my recommendation of the paper to a weak accept.
In particular the authors have added a number of clarification and additional experimental results that strengthen the paper compared to the original submission.

---

### Official Review · Reviewer_AiR8 · 2022-10-25

**Confidence:** 4
**Correctness:** 3
**Technical Novelty And Significance:** 3
**Empirical Novelty And Significance:** 3
**Recommendation:** 6

**Clarity, Quality, Novelty And Reproducibility:**

The paper is easy to read and understand. Authors present 3 problems and corresponding solutions clearly. There are a few novel ideas proposed with good intuitions. Looks it is reproducible upon the release of some key network modules.

**Details Of Ethics Concerns:**

Not aware of concerns.

**Strength And Weaknesses:**

+ Focus on more detailed local structures which are missing in the global layout input
+ SOTA results compared with existing counterparts, especially on the challenging COCO dataset
+ A novel semantic preservation module to reduce spatial resolution loss caused by generic CNNs

I just have a few following minor concerns:

How is the edge generator learned? Is it trained by some edge GT data or totally learned in an unsupervised way? More details about losses or designs are needed as it is one of the major contributions in this work

How many layout inputs are considered together when introducing the contrastive learning loss? How does this hyper-parameter affect the performance? Will the contrastive learning reduce diversity for a certain class that is by itself able to have multiple possible outputs? For example, for person class, it is likely to synthesize person in different clothing textures.

Which method in Figure 5 is the SOTA method?

Looks the user study in Table 1 is A/B test. Better to also report with the confidence intervals.

**Summary Of The Paper:**

This paper presents a semantic image synthesis method by generating edge as an intermediate state to guide the following generation. Meanwhile, a semantic preserving module is designed to selectively choose class-dependent feature maps. One contrastive learning strategy by considering different layout inputs is proposed to encourage generating similar content for same semantic class. The experiments show convincing qualitative and quantitative results to demonstrate the effectiveness of the proposed method.

**Summary Of The Review:**

This is an encouraging work that pushes the direction of more complex scene generation. The idea of contrastive learning by taking multiple layout into account is of some values.

---

### Official Review · Reviewer_UoWk · 2022-10-25

**Confidence:** 4
**Correctness:** 4
**Technical Novelty And Significance:** 4
**Empirical Novelty And Significance:** 3
**Recommendation:** 8

**Clarity, Quality, Novelty And Reproducibility:**

Given the listed strengths and weaknesses, the approach is novel and of high-quality. The presentation is clear. Because the proposed method is a complex system, which includes quite a lot of components. It seems to be difficult to reproduce.

**Strength And Weaknesses:**

Strengths:
- The proposed ideas leveraged additional labels such as edges and maximized the semantic layout label usage, which are interesting ideas. Jointly using various types of labels and proving those labels are helpful to boost performance are important contributions to peers and the community.
- Benchmark results consistently improve upon existing ones.
- Ablation studies are done comprehensively
- The motivation and approach are presented clearly and the paper is easy to read.

Weaknesses:
- The proposed approach leverages additional information, so it might not be fair comparison with existing methods. Given the scale of the training dataset, overfitting might exist. It would be interesting if authors could test the model on images in the wild or at least beyond the Cityscapes, ADE20K, and COCO-Stuff.


**Summary Of The Paper:**

This paper presents a ECGAN to advance semantic image synthesis. Three ideas were proposed in this paper to jointly boost the performance: the edge map with attention modules to guide image generation, the semantic preserving module together with the similarity loss on the semantic layout, and a pixel-wise contrastive learning mechanism. Experiments were done on commonly used datasets such as Cityscapes, ADE20K, and COCO-Stuff and output quality is measured by mIoU, Acc, FID and user studies. As shown in the paper, the proposed approach consistently outperforms existing benchmarks.

**Summary Of The Review:**

Overall, this paper presented a new SOTA for semantic image synthesis. The ideas are interning and the proposed system is novel. The evaluation is comprehensive.

---

### Decision · Program_Chairs · 2023-01-20

**Decision:**

Accept: poster

**Justification For Why Not Higher Score:**

The paper makes small contributions that allow to put together a well performing framework. Yet, the individual technical contributions are small, resulting in this paper being a borderline accept case. Therefore, we propose it for poster presentation.

**Justification For Why Not Lower Score:**

The technical contributions made in the paper are valid and yield good results in practice. Although the criticism regrding the paper layout, the paper is overall sufficiently well presented and clear.

**Metareview: Summary, Strengths And Weaknesses:**

This paper received two borderline reviews, one very positive review and one very critical review. Overall, the paper proposes a somewhat crafted framework for semantic image synthesis that presents several somewhat new ideas. The well engineered combination of these ideas allows to improve over the state-of-the-art in semantic image generation. The authors promise to publish the code upon acceptance.


**Note From Pc:**

if the above contains the word "oral" or "spotlight" please see: "oral" presentation means -> notable-top-5% and "spotlight" means -> notable-top-25%. As stated in our emails, we are disassociating presentation type from AC recommendations

**Summary Of Ac-Reviewer Meeting:**

In the virtual meeting, the contributions of the paper were discussed among the AC and and the three lower scoring reviewers. As a result, all reviewers and the AC agree that the proposed ideas and the way they are put togehter qualifies for accpetance at ICLR.
At the same time, we also noticed that the paper is not particularly well written / polished. The revision addressed some of the original concerns regarding notation/clarity and presented results. Yet, paper layout is not appealing after the revision, with tables being very small and spaces being too small. We strongly advice the authors to address these issues for their final submission.